# Satellite Video Moving Vehicle Detection and Tracking Based on Spatiotemporal Characteristics

**DOI:** 10.3390/s23125771

**Published:** 2023-06-20

**Authors:** Ming Li, Dazhao Fan, Yang Dong, Dongzi Li

**Affiliations:** Institute of Geospatial Information, Information Engineering University, 62 Science Avenue, Zhengzhou 450001, China; liming12102022@163.com (M.L.); dongyang33@aliyun.com (Y.D.); li_5588_dong@163.com (D.L.)

**Keywords:** spatiotemporal characteristics constraint, satellite video, moving vehicle detection, data association

## Abstract

The complex backgrounds of satellite videos and serious interference from noise and pseudo-motion targets make it difficult to detect and track moving vehicles. Recently, researchers have proposed road-based constraints to remove background interference and achieve highly accurate detection and tracking. However, existing methods for constructing road constraints suffer from poor stability, low arithmetic performance, leakage, and error detection. In response, this study proposes a method for detecting and tracking moving vehicles in satellite videos based on the constraints from spatiotemporal characteristics (DTSTC), fusing road masks from the spatial domain with motion heat maps from the temporal domain. The detection precision is enhanced by increasing the contrast in the constrained area to accurately detect moving vehicles. Vehicle tracking is achieved by completing an inter-frame vehicle association using position and historical movement information. The method was tested at various stages, and the results show that the proposed method outperformed the traditional method in constructing constraints, correct detection rate, false detection rate, and missed detection rate. The tracking phase performed well in identity retention capability and tracking accuracy. Therefore, DTSTC is robust for detecting moving vehicles in satellite videos.

## 1. Introduction

In recent years, low-orbiting video satellites have gradually become a frontier technology in the development of remote sensing because of their ability to quickly acquire high-resolution video data over large areas [1]. Compared to traditional optical remote-sensing satellites, the most important feature of video satellites is that they can acquire dynamic information concerning the Earth’s surface. Rich information on vehicle dynamics in satellite videos is significant for urban traffic planning, road construction, and smart city buildings. Therefore, the detection and tracking of moving vehicles using satellite video have become a current research hotspot in remote sensing [2].

Compared with normal ground-based videos, satellite videos must consider their unique characteristics when detecting and tracking moving vehicles. The problems of scale variation and inter-target occlusion faced by ground-based videos have been addressed less by satellite videos [3]. However, the high-altitude nature of video satellites allows for a wide range of images, resulting in backgrounds containing many complex features and moving targets that appear faint and tiny. Simultaneously, video satellite gaze-imaging properties cause stationary targets to appear in motion between frames, that is, parallax pseudo-motion [4]. Consequently, traditional motion target detection algorithms cannot effectively identify true motion targets, pseudo-motion targets, and noise. In addition, in the tracking phase, the vehicle identity cannot be effectively identified because of similar vehicle characteristics. Therefore, accurate detection and tracking of moving vehicles from satellite videos remains a difficult task [5].

For this reason, many scholars have proposed the use of prior road knowledge to optimize target detection algorithms [6,7,8,9]. Detection precision has been significantly improved by creating a road mask to reject a large amount of noise and pseudo-motion interference from the background. However, this approach faces various problems in establishing constraints. Methods based on the accumulation of motion information use time domain information. This type of algorithm has poor stability, cannot effectively accumulate complete masks when traffic volumes are low, and requires a significant amount of time prior to detection. Methods based on single-frame object-oriented classification use spatial domain information. This type of method is computationally complex and suffers from problems such as road misses and breakages. Methods based on loading existing road network information utilize existing results. This method requires considerable storage space and the manual creation of road network data in advance.

During the tracking phase, the focus is on correlating the detected targets between frames. Previously, data association was performed by constructing motion features [10,11,12]. With the rapid development of neural networks, methods based on depth features have gradually become mainstream [13,14]. However, the small size, large number, and similarity of the features of the moving vehicles in the satellite video make consistently tracking the vehicles between frames using only depth features difficult.

Previous research has clearly shown that most algorithms suffer from three problems: ① Most algorithms construct a priori constraints considering only time domain or spatial domain information and do not fully combine the two, resulting in poor algorithm stability. ② Most algorithms are less effective in dealing with faint and small moving targets. ③ During the tracking phase, most algorithms rely excessively on deep features. Therefore, this study proposed a satellite video moving vehicle detection and tracking method based on spatiotemporal characteristic constraints called Detection and Tracking Based on Spatiotemporal Characteristics (DTSTC).

The remainder of this paper is organized as follows: Section 2 provides an overview of the existing research relevant to this study. Section 3 presents the implementation details of the proposed moving vehicle detection and tracking method. Section 4 presents the experimental results obtained using the proposed method at various stages. Finally, Section 5 summarizes the methodology.

## 2. Related Work

### 2.1. Traditional Background Subtraction

Traditional motion target detection can be divided into four categories: temporal difference, optical flow, background subtraction, and deep-learning-based methods [15]. The temporal difference method is a simple low-volume algorithm; however, it cannot adapt to complex environments or overcome the void phenomenon. Optical flow methods are computationally complex and are sensitive to light. Deep-learning-based methods are computationally complex and data sensitive. In contrast, background subtraction is rich in variety, and it is more effective [16]. KaewTraKulPong and Bowden proposed modeling the change in a particular pixel over time using multiple Gaussian models for the background [17]. This method helps restore the target profile to its original shape, but the detection results are not excellent. The visual background extractor (ViBE) algorithm proposed by Barnich and Van Droogenbroec builds a background model by aggregating the previous observations at each pixel location [18]. However, this method suffers from ghosting, incomplete target extraction, and a shadowed foreground. Liu and Zhang proposed an improved ViBe algorithm. In order to reduce the repetition rate of the pixel value in the background model, the proposed method changes the neighborhood selection method to improve the accuracy of the initialization of the background model [19]. A detection framework called E-GMMTFD was proposed by Shu et al. [20]. The E-GMMTFD framework utilizes the adaptability of the GMM to complex environmental changes and the strength of the TFD method to eliminate background movement, thus reducing the false alarms caused by illumination change and background movement. However, traditional background subtraction methods do not adequately consider the characteristics of satellite videos and cannot effectively distinguish between noise interference, pseudo-motion targets, and moving vehicles.

### 2.2. Detection Methods Based on Road Constraints

Using prior road knowledge to optimize the target detection algorithm is the easiest and most effective way to remove background noise and pseudo-motion target interference. Kang et al. detected moving target regions using temporal differences, and after accumulating the moving regions to generate regions of interest, moving vehicles were detected [6]. Lei et al. used the boundary association of superpixels combined with background subtraction to generate a motion heat map, thereby completing superpixel-based motion region estimation and improving target detection results [7]. However, this method requires a large number of moving vehicles and relies on the accumulation of many video frames, which is a significant waste of video frame resources. Lu et al. used an object-oriented classification approach to extract road masks, which were then used to reject false motions and pseudo-motion targets using the ViBE algorithm [8]. However, this method uses only a single image frame, which can result in significant breaks and missed detections. Yin directly used existing road network data overlaid with satellite video data [9]. When solving problems such as road breaks and missed inspections, the global operating range of video satellites can lead to an overwhelming amount of data on the road network.

### 2.3. Data Association

Current inter-frame target association methods often use target motion and appearance features. Bewley et al. used Kalman filtering (KF) [21] to calculate the prediction frame with the detection frame obtained by Faster R-CNN [22] for intersection over union (IOU) calculations to associate the target [10]. Zhou used convolutional networks to model movement patterns and interaction relationships [11]. Shan et al. used a graph convolutional network to fuse multiframe image information for target location prediction [12]. This method causes significant degradation in tracking performance in dense scenes because it uses only motion information. Yu et al. extracted target appearance features using GoogLeNet and used the k-nearest neighbor algorithm to associate targets [13]. Lee and Kim enhanced the target discrimination by fusing feature pyramids, thus enabling inter-frame target associations [14]. This method is prone to problems, such as tracking frame drift in the face of the small size and similarity of features of the moving vehicles in the satellite video. Wang et al. proposed a graph neural network for joint multi-target tracking with an optimized detection and tracking phase [23]. Lit et al. proposed a self-correcting KF position prediction algorithm combined with recurrent neural networks for inter-frame target association [24]. This method is usually slow to track, owing to the high complexity and computational effort of the network. In summary, existing algorithms do not exhibit good applicability in the case of small sizes, large numbers, and similar features of moving vehicles in satellite videos.

## 3. Methods

DTSTC, as proposed in this study, consists of four parts: video data pretreatment, construction of spatiotemporal constraints, moving vehicle detection, and inter-frame vehicle association (Figure 1). The original satellite video data were preprocessed with intra-frame denoising and inter-frame image stabilization. Road masks containing spatial information were then extracted within fixed-interval frames using the D-LinkNet, whereas a motion heat map containing temporal information was cumulatively generated using temporal differences. The two are fused to generate a region-of-interest constraint and improve the precision of moving vehicle detection. In addition, a local contrast enhancement algorithm based on the LAB color space [25] was used to optimize the k-nearest neighbor (KNN) moving target detection in the region of interest, resulting in the greatly improved detection of moving vehicles. Finally, vehicle tracking was performed using improved data association methods for the inter-frame association of moving vehicles.

### 3.1. Constraint Construction Based on the Spatiotemporal Domain

Among the existing constraint construction methods, the motion-based accumulation method is unstable and requires the accumulation of a large number of video frames. Single-frame object-oriented classification methods for extracting road masks are prone to breakage and missing detections. The method for loading existing road maps has a high memory footprint and requires significant manual labeling. Therefore, DTSTC incorporates road masks and motion heat maps to improve the robustness of the constructed constraints. The road mask in this method fully utilizes spatial domain information and can provide an improved initial value for the motion heat map. The motion heat map uses time domain information, which complements the broken and missed detection parts of the road mask. In addition, this method has a small memory footprint compared to loading an existing road network and does not require extensive manual marking.

The D-LinkNet road extraction fully considers the spatial representation characteristics of roads in a single image frame, including the narrowness, connectivity, complexity, and long spans. The network also exponentially increases the perceptual field of the model using a jump connection and null convolution module, making it more capable of road extraction. Then, ResNet34 [26] is used as the encoder to effectively solve the problem of losing image feature information during the convolution process. The network exhibited superior results in terms of road extraction when faced with satellite video data. Therefore, DTSTC used the D-LinkNet for road extraction from a single image frame to initialize the constraints.

However, D-LinkNet only extracts the road mask using the spatial characteristics of a single frame of an image, does not take full advantage of the video data in the time domain, and cannot avoid the occurrence of broken and missing roads. Therefore, DTSTC complemented this by cumulatively generating a motion heat map using temporal differences. This temporal difference fully utilizes the target motion information between two successive frames and can effectively complement the road mask generated using only a single image frame. Figure 2 illustrates the general flow for generating a coarse-motion heat map using temporal differences. The video frames are first detected quickly using the temporal difference, which has a high detection efficiency. The results of each frame are then processed using mathematical morphology to remove the effects of random noise and pseudo-motion targets. Finally, the processing results were overlain to produce an approximate motion heat map.

The specific integration of the two is shown in Figure 3. The connected domain of the rough motion heat map was first preprocessed and filtered based on parameters such as the area (A), aspect ratio ($A_{r}$), perimeter-to-area ratio ($R_{ca}$), and surface roughness ($S_{r}$) to remove the influence of pseudo-motion targets. The aspect ratio and surface roughness were calculated using Equations (1) and (2), respectively:(1)$$A_{r}=l\;/\;w,$$
(2)$$S_{r}=C_{mech}\;/\;C_{a},$$
where l denotes the minimum external rectangular length of the connected domain, and w denotes the minimum external rectangular width of the connected domain. $C_{mech}$ is the minimum external convex envelope perimeter of the connected domain, and $C_{a}$ is the connected domain perimeter.

The preprocessed motion heat map was then graphically compared with the road mask, and the remaining connected domains in the motion heat map were conditionally discriminated. If these requirements were met, the domain was included in the road mask. Otherwise, the connected domains were discarded. The goodness of the discriminant condition directly determines the effectiveness and stability of the constraint construction. We combined the topological relationship between the two and the growth rate of the motion heat map and set the discriminant condition as follows: ① The motion heat map connectivity domain intersects the road mask. ② The historical aspect ratio and length growth rate of the motion heat map simultaneously satisfy the corresponding thresholds. One of these two conditions must be satisfied. Condition ① effectively raises the priority of motion targets around the road and retains their cumulative results, while suppressing them for motion targets separate from the road area. The length growth rate under condition ② can effectively identify the connected domain formed by noise and moving vehicles. The historical aspect ratio represents the aspect ratio of the connected domain during the motion accumulation process, which eliminates the effect of a surge in the length of the connected domain caused by large noise in one frame during the motion accumulation process. As shown in Figure 4, connected domains *B* and *C* both intersect the road mask and satisfy condition ①. The length growth rate and historical aspect ratio of connected domain *A* meet the threshold requirements and qualify for condition ②. In contrast, connected domain D does not meet either condition, and it is selected for elimination. The specific discriminant condition is described by Equation (3):(3)$$I_{d}{UI}_{i}^{\prime }>0\;\mathrm{or}\;{\tilde{A}}_{r}^{i}>T_{A}\;\mathrm{and}\;R\left(l_{i}\right)>T_{R}\;\left(d=1,2,\cdot \cdot \cdot ,m;\;i=1,2,\cdot \cdot \cdot ,n\right)$$
where $I_{d}$ denotes the *d*-th connected domain in the road mask, $I_{i}^{\prime }$ denotes the *i*-th connected domain in the motion heat map, ${\tilde{A}}_{r}^{i}$ denotes the historical aspect ratio of the *i*-th connected domain in the motion heat map, R denotes the formula for calculating the growth rate of the length of the connected domain, $T_{A}$ denotes the connected domain aspect ratio threshold, and $T_{R}$ denotes the connected domain length growth rate threshold; m is the number of connected domains in the road mask, and n is the number of connected domains in the motion heat map.

### 3.2. Moving Vehicle Detection

Moving vehicles in satellite videos are presented as faint and tiny targets, and direct detection using traditional background subtraction can result in serious under-detection. To compensate for vehicles being difficult to distinguish from the background owing to their low contrast, DTSTC optimized the KNN moving vehicle detection algorithm using the LAB color-space local contrast enhancement algorithm.

Owing to the variety of colors in the background areas, the effect can be significantly reduced or even have side effects when contrast enhancement is applied to the image as a whole. We begin with the contrast enhancement within the constrained region constructed in the previous section. Specifically, the mean $I_{\mu }={\left[L_{\mu },a_{\mu },b_{\mu }\right]}^{T}$ of the luminance (L) and color (a, b) feature components of the LAB model are extracted within the constrained region, as shown in Equation (4):(4)$$I_{\mu }=\left[\begin{array}{c} L_{\mu }\\ a_{\mu }\\ b_{\mu } \end{array}\right]=\frac{1}{W\times H}\left[\begin{array}{c} \sum_{i=1}^{W}\sum_{j=1}^{H}L(u,v)\\ \sum_{i=1}^{W}\sum_{j=1}^{H}a(u,v)\\ \sum_{i=1}^{W}\sum_{j=1}^{H}b(u,v) \end{array}\right].$$

Subsequently, a Gaussian low-pass filter is applied to the image to obtain the filtered image $I_{g}$, as shown in Equation (5):(5)$$I_{g}=I(u,v)⊗G(x,y,\sigma ).$$

The low-pass filtered image is converted to the LAB color space, and the *L*, *a*, and *b* channels are separated to obtain the three features, $I_{\omega hc}$, as shown in Equation (6):(6)$$I_{\omega hc}={\left[L_{g},a_{g},b_{g}\right]}^{T}.$$

These features are fused to obtain a significant image value. The resulting highlighted moving vehicles can significantly improve the completeness of moving vehicle detection. Based on this, the KNN motion target detection algorithm was used to detect areas within the constraints, resulting in substantially improved detection results.

### 3.3. Inter-Frame Vehicle Association

The moving vehicles in the satellite video occupied only a few pixels and had similar shapes, colors, and other features. This results in many false matches when matching vehicles between frames using only features such as appearance and color. Vehicle motion in satellite videos can be approximated as a two-dimensional planar motion; therefore, the two-dimensional position information of moving vehicles in satellite videos is more useful for describing inter-frame vehicle association. Based on the detection of moving vehicles, this study proposed a vehicle association method that uses a combination of the relative vehicle position and historical motion information.

The moving vehicle is first defined as a vector with position and velocity using a Kalman filter. The covariance matrix infers the next frame state, that is, the trajectory position of the moving vehicle, from the current frame state of the target. The Kalman filter prediction equation is given in Equation (7):(7)$$\left\{\begin{array}{c} {\hat{x}}_{k}=A{\hat{x}}_{k-1}\\ {\bar{P}}_{k}=AP_{k-1}A^{T}+Q \end{array}\right.$$
where ${\hat{x}}_{k}$ denotes the target kth frame state volume, A is the state transfer matrix, P is the state covariance matrix, and Q is the process noise covariance matrix. After associating the next frame detection result, the target state is updated, and the update equation is shown in Equation (8):(8)$$\left\{\begin{array}{c} K_{k}={\bar{P}}_{k}H^{T}{\left(H{\bar{P}}_{k}H^{T}+R\right)}^{-1}\\ {\hat{x}}_{k}={\hat{x}}_{k}+K_{k}\left(z_{k}-H{\overset{-}{\hat{x}}}_{k}\right)\\ P_{k}=\left(I-K_{k}H\right){\bar{P}}_{k} \end{array}\right.$$
where $z_{k}$ is the state-value observation vector, H is the state-observation matrix, R is the measurement noise covariance matrix, K is the Kalman gain, and I is the unit matrix.

As the video contains multiple targets, data association is required to match the predicted results with the detection results of the next frame to enable vehicle tracking. For frame t+1, m moving targets can be detected using the detection method described in Section 3.2, and the Kalman filter predicts the target position for frame t+1 based on the tracking results for frame t, yielding n predicted targets, to form a data correlation matrix as shown in Equation (9):(9)$$C=\left[\begin{array}{cccc} c_{11} & c_{12} & \cdots & c_{1n}\\ c_{21} & c_{22} & \cdots & c_{2n}\\ \vdots & \vdots & \ddots & \vdots \\ c_{m1} & c_{m2} & \cdots & c_{mn} \end{array}\right],$$
where C is the data association matrix and $c_{mn}$ denotes the match between the *m*-th detection and *n*-th prediction frames. Finally, the Hungarian algorithm was used to obtain the optimal solution for the data association matrix to complete the inter-frame moving vehicle association.

The degree of matching is a measure of the similarity of vehicles between frames; therefore, how well the degree of matching is constructed determines the success of the association of vehicles between frames. We fully considered the characteristics of the moving vehicles in the satellite video, integrated the features such as position and motion topology relationship, and proposed a matching degree construction method combining the intersection and merging ratio with the angular difference of motion direction. The IOU effectively describes the position relationship between the target detection and prediction frames, and the angular difference in direction of motion fully utilizes the historical position information of the target for its evaluation. As shown in Figure 5, when the green target prediction frame intersects the two detection frames with the same IOU, the effect of the wrong target can be effectively rejected using angles $\alpha_{1}$ and $\alpha_{2}$. The linear combination of the two allows for a more robust description of the inter-frame vehicle matching similarity. The matching degree construction equation is given as Equation (10):(10)$$C_{\mathrm{ij}}=\beta C_{diou}\left(i,j\right)+\varepsilon C_{top}\left(i,j\right),$$
where $C_{diou}$ is the IOU of the two; $C_{top}=\cos\left(\alpha \right)$ is the cosine of the directions of the target prediction and detection frames; α is the angle between the t detection frame relative to the t+1 detection and prediction frames, as shown in Figure 5; and β and ε are weighting factors that sum to 1.

## 4. Results

### 4.1. Test Data

Six datasets from VISO [27] were selected to verify the validity and practical performance of the method. The dataset was captured using the Jilin-1 satellite constellation. The spatial resolution of the Jilin-1 video satellite is 1.03 m, with an altitude of 535 km, an orbital inclination of 97.54°, a standard view area of 11 km × 4.5 km, and an uncontrolled positioning accuracy of approximately 200 m. The data include an RGB color video, and the dataset is created by intercepting standard images. The videos cover several square kilometers in real scenes. Figure 6 shows the first video frame for each area. The video was taken in St. Paul, USA, and Muharraq, Bahrain. For each scene, 100 frames were used for testing. The video includes not only roads but also residential land, wood, arable land, water bodies, and other feature types. Specifically, Area 1 comprises a main road with a large area of rural residential land and vegetation in the background. Area 2 comprises a main road with a large area of a lake in the background. Area 3 comprises mainly an overpass with a large area of residential land in the background. Area 4 comprises a road and a path with a town settlement in the background. Area 5 consists of a crossroad with a town settlement in the background. Area 6 comprises the coastal highway with a large body of water in the background. The above 6 are sufficient data for a more comprehensive assessment of the actual performance of DTSTC.

### 4.2. Evaluation Indicators

To quantify the accuracy of the target detection algorithm and compare it with previous algorithms, the accuracy evaluation metrics used were the precision (*P*), recall (*r*), and F1 score (*F*), which are commonly used in the target detection field. This was calculated using Equation (11):(11)$$\begin{array}{c} P=\frac{TP}{TP+FP}\times 100\%\\ r=\frac{TP}{TP+FN}\times 100\%\\ F=2\times \frac{P\times r}{P+r}\times 100\% \end{array},$$
where TP denotes the number of correct motion target detections, FP denotes the number of incorrect motion target detections, and FN denotes the number of undetected motion targets.

For the inter-frame association of moving vehicles, we used the number of correct tracking target identity detections (*IDTP*), number of false tracking target identity detections (*IDFP*), number of missed tracking target detections (*IDFN*), multiple object tracking accuracy (*MOTA*), total number of object identity switches (*IDS*), and identity F1 score (*IDF*_1_). To obtain accurate precision results, the *IDP*, *IDP*, *IDF*, and *IDS* were obtained by manual counting. Among all the evaluation metrics, *MOTA* and *IDF*_1_ are the most important for evaluating multi-objective tracking algorithms, as shown in Equations (12) and (13), respectively:(12)$$MOTA=1-\frac{\sum \left(FN+FP+IDS\right)}{\sum GT},$$
(13)$$IDF_{1}=\frac{2IDTP}{2IDTP+IDFP+IDFN},$$
where GT indicates the number of true bounding boxes.

### 4.3. Constraint Construction Validation Tests

To verify the effectiveness of DTSTC for constructing the spatiotemporal domain constraints, the temporal difference cumulative motion heat map method [6], U-Net road extraction algorithm [28], and D-LinkNet road extraction algorithm [29] were compared with DTSTC. The method in this study used the threshold settings of $T_{A}$ = 5 and $T_{R}$ = 0.3. The results of the comparison tests are presented in Figure 7. As can be seen from the results, the motion heat map accumulated using the temporal difference is less effective, with no superior road masks obtained in the six trial sets and large non-road areas accumulated in both Areas 2 and 3. This is because there were fewer vehicles in the video frame, and it was not possible to effectively accumulate sufficient thermal areas of motion in a few frames to form a road mask. In addition, noise accumulated in the video, creating pseudo-motion heatmap areas. Therefore, this method is only suitable for city centers and areas with high traffic volumes. The U-Net road extraction algorithm extracted only relatively complete road masks in Areas 4 and 5, with incorrect extraction in all other areas. This is because the U-Net road extraction network has a small feature perception field and does not fully consider the complex structural characteristics of roads; therefore, there are obvious omissions and breakdowns. D-LinkNet largely ensures the integrity of the extraction results. However, road breaks still exist, such as in the upper-right trail in Area 3 and the lower-middle bypass in Area 6. DTSTC fully utilizes the time domain information to complement the road mask extracted by D-LinkNet and achieves better results. Most of the road area was extracted in all six trial sets, and there was improvement in breakage and missed detections that existed with D-LinkNet. Breaks and misses in the red areas in the Figure 7 have been effectively addressed. Therefore, the proposed algorithm can build constraints more robustly and provide a good basis for the subsequent detection of moving vehicles.

To verify the effect of stepwise optimization of the mask generation method in this study, a set of video data was selected for experimentation. The D-LinkNet mask was supplemented every 15 frames using the motion heat map generated by the frame difference method, as illustrated in Figure 8, which shows the effect of mask fusion at each stage. As shown in the diagram, the missing roads are gradually supplemented and do not cause excessive noise. The feasibility and superiority of DTSTC in constructing constraints based on the spatiotemporal domain are demonstrated by the progressive addition of road network masks.

### 4.4. Comparative Testing and Quantitative Analysis of Moving Vehicle Inspection

To verify the effectiveness of the proposed method in the moving vehicle detection phase, several of the best motion target detection algorithms developed in recent years were selected for comparison tests on the experimental data. The comparison methods include temporal difference (TD), an improved mixture of Gaussian v2 (MOGv2), ViBE, the MOGv2 algorithm based on the constraints in this study (CMOGv2), the method presented in Literature 6 (MVDSV), and the method presented in Literature 8 (VOMVD).

Figure 9 shows the detection results for each algorithm in Area 1. The results show that the temporal difference detects most of the moving vehicles but is severely affected by noise and produces many false detections owing to minor changes in the background, as shown in Figure 9b-6,b-7. There are also serious “double shadows,” as shown in Figure 9b-5. There are incomplete detection frames when MOGv2 detects larger targets, as shown in Figure 9c-5, and there are also false detections due to background variations, as shown in Figure 9c-6,c-7. ViBE moving target detection was heavily ghosted, as shown in Figure 9d-5, and was unable to suppress the effects of background noise, as shown in Figure 9d-7. CMOGv2 effectively eliminated noise and pseudo-motion interference in the background, as shown in Figure 9e-7. However, the phenomenon of missed detection remains, and the phenomena of “double shadow” and “ghostly shadows” occur for large targets without texture, as shown in Figure 9e-6. Although MVDSV partly removes the interference of noise in the background, as shown in Figure 9f-7, there are missed detections, as shown in Figure 9f-5,f-6. The VOMVD detection target is the presence of ghosting, as shown in Figure 9g-5, as well as missed detection (Figure 9g-6).

In contrast, DTSTC eliminates the influence of pseudo-motion targets and background noise with very few missed detections, as shown in Figure 9h-7. There are no “double shadows” for larger targets, as shown in Figure 9h-5,h-6. Therefore, DTSTC has a significant advantage over traditional methods in terms of visualization.

To quantify and analyze the detection results, *P*, *r*, and F1 score were calculated separately for each algorithm within each area. As shown in Table 1, bold text indicates optimal values and underlined text indicates suboptimal values. Within Area 1, DTSTC achieves the highest detection precision. The recall of this method was slightly lower than that of temporal difference. The analysis suggests that this may be due to the low threshold setting for the inter-frame differencing method, which increases the detection recall at the expense of lower detection precision (no distinction between noise and moving targets).

Within Area 2, DTSTC had the highest recall and F1 scores. The accuracy was slightly lower than that of VOMVD; however, the recall was considerably higher than that of VOMVD. Comparison tests show that the accuracy of the method with constraints is considerably higher than that of the conventional method, which is analyzed because the background of Area 2 contains a large lake, and the diffuse reflection of the lake causes a lot of noise interference. The constraint effectively eliminates this type of background interference.

The best results in terms of precision, recall, and F1 score were achieved in Areas 3–5. This is because there is less noise interference in these areas and a greater degree of road and vehicle disparity, providing better initial conditions for moving target detection. Spatiotemporal domain constraints build complete and contrast-enhanced assisted detection, greatly enhancing detection capabilities and enabling the robust detection of vehicles in motion satellite videos.

Within Area 6, both DTSTC and CMOGv2 performed well. DTSTC was slightly less accurate than CMOGv2 in terms of detection precision. The reasons for this are as follows: On the one hand, it may be that the MOGv2 misdetection phenomenon causes it to have no misdetection noise; therefore, the phenomenon of low recall and high precision occurs. On the other hand, it may be that the contrast enhancement algorithm in DTSTC increased the influence of noise within the road, resulting in more mis-detected targets.

Figure 10 and Figure 11 provide statistics on the detection precision and recall of each algorithm for different areas. As shown in the chart, DTSTC has the best detection accuracy in most cases, it being only slightly lower than CMOGv2 in Area 6. The recall rates were low in Area 2 under the influence of harsh environmental factors, while the rest of the area maintained high levels.

To further validate the stability of the algorithm, Figure 12 shows the variations in the precision and recall for each area over 100 frames. The detection precision of DTSTC remained high in all areas and varied smoothly between the frames. There are more noticeable fluctuations in the recall, but they still perform better than the other methods. Therefore, DTSTC is not only better in terms of average detection performance, but also exhibits good results in terms of detection stability.

By combining the results of each area detection for the satellite video data, we find that the temporal difference and MOGv2, despite maintaining a good recall rate, are subject to serious interference and have a low precision rate. Although ViBE performs more consistently, the overall precision is low, and there is significant “ghosting” and missed detections. CMOGv2 showed significant improvement in precision but still had a high rate of missed detections. Although MVDSV improves detection accuracy through heat map constraints, the recall performance is mediocre, which is probably due to the instability of the constraint construction when analyzed. The VOMVD performed well overall, but there is still room for improvement in the recall rate. In contrast, DTSTC achieved better results in terms of both detection precision and recall, with the highest F1 score. Therefore, the detection algorithm in DTSTC can also perform well in the face of the complex background and noise of satellite video data and has certain advantages in terms of robustness and applicability.

### 4.5. Moving Vehicle Tracking Test

To verify the effectiveness of DTSTC during the tracking phase, it was tested for four scenarios. The visualization results are shown in Figure 13, with different colored boxes representing different tracked objects. Figure 13a shows the results of long-distance tracking for a single vehicle. The blue box in the diagram tracks the target with good results over long distances, no ID switching occurs during tracking, and the tracking box position is more accurate. Figure 13b shows the vehicle tracking results obtained using a right-angle bend. The red box tracks the target steadily through the bend without ID switching, and the vehicle color remains similar to the background without losing the target. Figure 13c shows the more intensive multivehicle tracking results, where the proposed algorithm tracks stably, with no significant false detections, missed detections, or ID switching. Figure 13d shows the vehicle tracking results with partial occlusion. Although the tree branches occluded the vehicle, the tracking results were good, with no ID switching or missed detections. Thus, the proposed tracking algorithm has a more robust tracking performance in complex environments such as long sequences, non-linear, dense multiple targets, and partial occlusion.

A small area within each area was intercepted for testing to further verify the effectiveness of the data association method. The experiments compared the CMOGv2 approach combined with the data association approach (CMOGv2_D), the detection approach combined with the SORT algorithm (C_SORT), and DTSTC. In Table 2, the optimal values are indicated in bold. The proposed method is more stable in most areas, multitarget tracking precision remains high, tracker identity maintenance is excellent, and the number of ID switches is low. The method of tracking data associations using SORT was the next most effective, maintaining a high overall level. However, the method using CMOGv2 detection was less effective. Notable among these is the low precision of multitarget tracking for each method in Area 2. This is because missed detections occurred during the detection phase, resulting in certain targets not being tracked during the tracking phase. In Area 3, the results of the proposed method and the method of tracking data associations using SORT were free of omissions and misdetections. This is because there is less traffic in the area, vehicle contrast is evident, and excellent detection provides good initial values for tracking.

In summary, DTSTC used in this study can effectively correlate inter-frame vehicles in satellite videos. However, tracking precision largely depends on the detection effect. Although the detection method in DTSTC has largely improved the detection precision and recall rate, the phenomenon of missed detection still exists, which leads to ID switching or missed targets in the tracking targets.

## 5. Conclusions

Given the complex background of satellite videos, noise, and pseudo-motion targets that interfere with the detection and tracking of moving vehicles, this study proposed a method for detecting and tracking moving vehicles in satellite videos with spatiotemporal characteristic constraints (DTSTC). Compared with traditional methods, DTSTC is more robust in constructing constraints and removing background noise interference. The detection rate of moving target extraction was improved, and a significant advantage was achieved in detecting moving vehicles. DTSTC was also proven to be more stable during the tracking phase.

Although DTSTC reduced the rate of missed detection of moving vehicles to a certain extent in this study, there were still missed detections of faint targets. To further improve detection precision, the incorporation of time-dimensional motion information can be considered in the detection process. In addition to the method proposed in this study, as with most current algorithms, the tracking performance depends on the detection algorithm. In the future, further consideration should be given to transferring features from the past moments of the target to the current frame, enhancing detection; thus, detection and tracking can reinforce each other and form a closed loop.

## Figures and Tables

**Figure 1 sensors-23-05771-f001:**
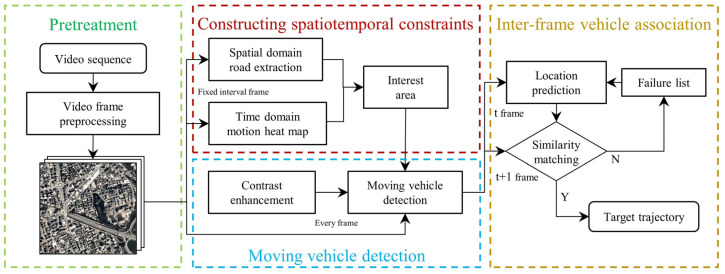
DTSTC flow diagram. The video frame data is processed in the order of left to right to obtain the tracking results. The input is the satellite video data, while the output is the tracking track result of the moving vehicle in image coordinates.

**Figure 2 sensors-23-05771-f002:**
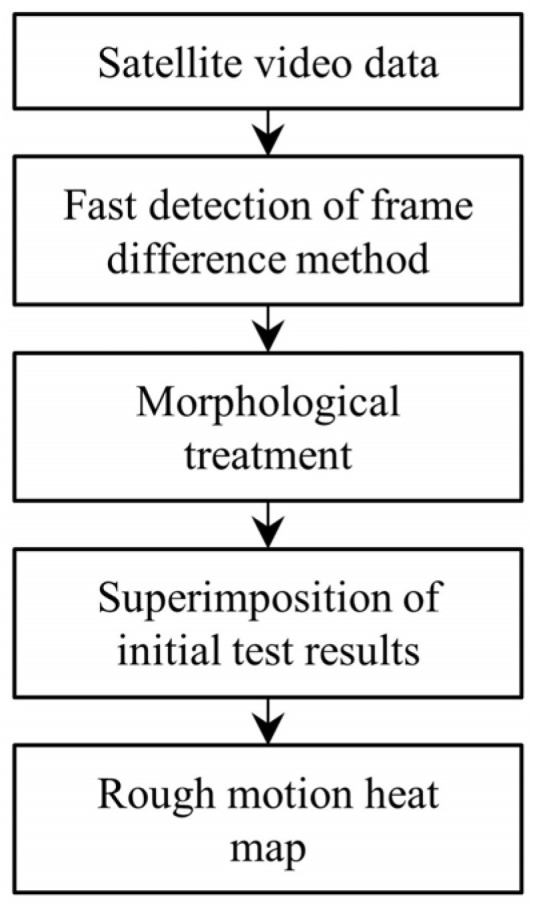
Flow chart depicting the automatic acquisition technique for the approximate motion heat maps.

**Figure 3 sensors-23-05771-f003:**
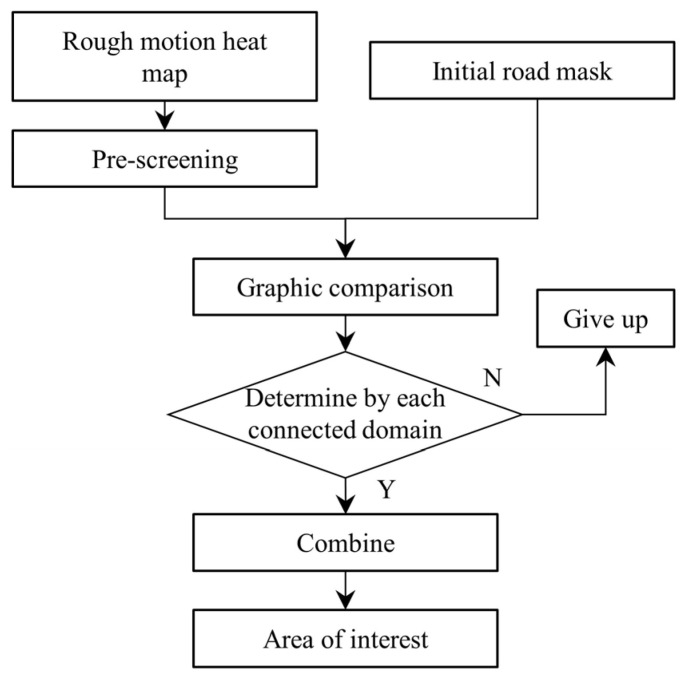
Flowchart of initial road mask and rough motion heat map fusion.

**Figure 4 sensors-23-05771-f004:**
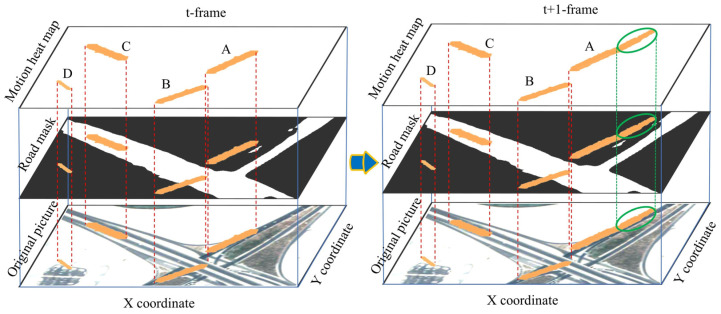
Schematic diagram of the discriminant condition. The left diagram shows frame *t*, and the right diagram shows frame *t* + 1. The bottom layer of the diagram represents the original image, the middle layer represents the D-LinkNet extracted road mask, and the top layer is the motion heat map connectivity domain area. The area circled in green is the connectivity domain for inter-frame variations.

**Figure 5 sensors-23-05771-f005:**
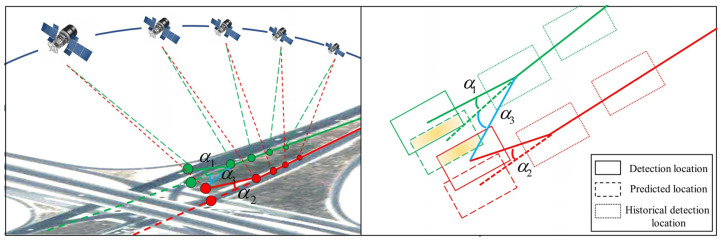
Diagram of the angular difference in the direction of motion. The green and red dots represent the positions of the two moving vehicles in the video in multiple frames, with the solid line indicating their actual trajectory, and the dashed line indicating the predicted trajectory. $\alpha_{1}$ indicates the angle between the green vehicle detection and predicted targets, $\alpha_{2}$ indicates the angle between the red vehicle prediction and detection targets, and $\alpha_{3}$ indicates the angle between the green vehicle prediction and red vehicle detection targets.

**Figure 6 sensors-23-05771-f006:**
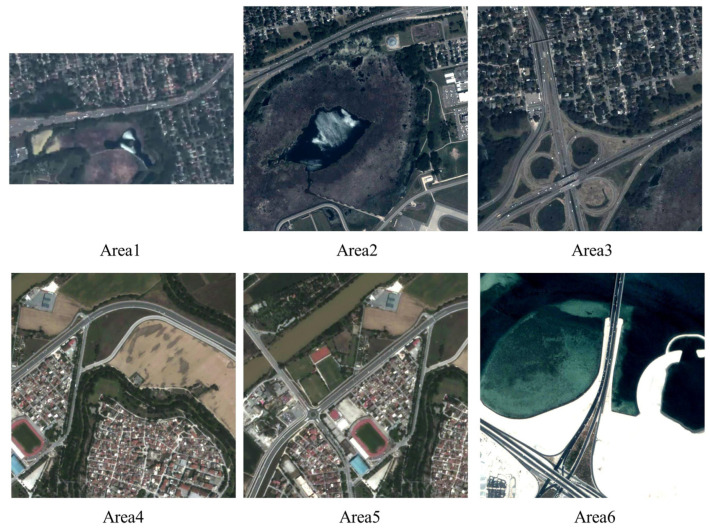
The first frame images of video data.

**Figure 7 sensors-23-05771-f007:**
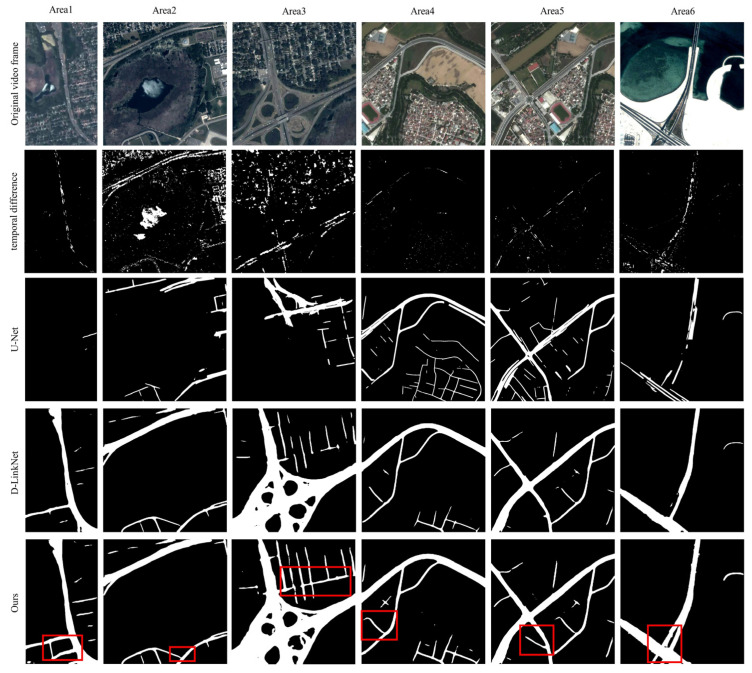
Comparison results of the extraction of the areas of interest.

**Figure 8 sensors-23-05771-f008:**
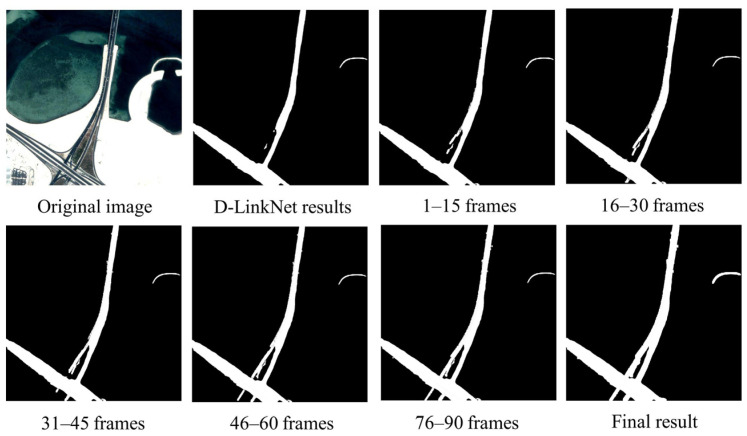
Results of the stepwise optimization of the mask.

**Figure 9 sensors-23-05771-f009:**
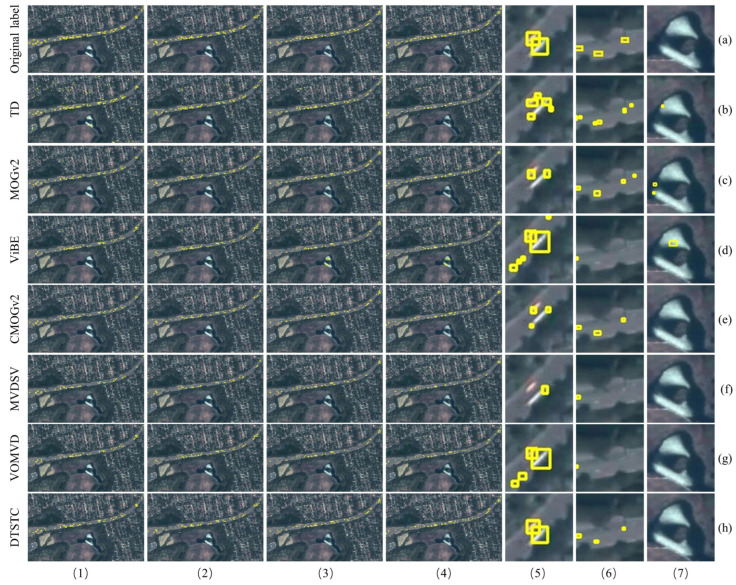
Comparison diagram for Area 1 moving vehicle detection, where columns (**1**)–(**4**) are the detection results in Area 1 for frames 30, 50, 70, and 82, respectively. Columns (**5**) and (**6**) show the road area refinement maps. Column (**7**) shows the refinement of the background area. The yellow box in the diagram shows the detected moving vehicles. For ease of description in the subsequent text, the names of the methods are replaced by (**a**–**h**).

**Figure 10 sensors-23-05771-f010:**
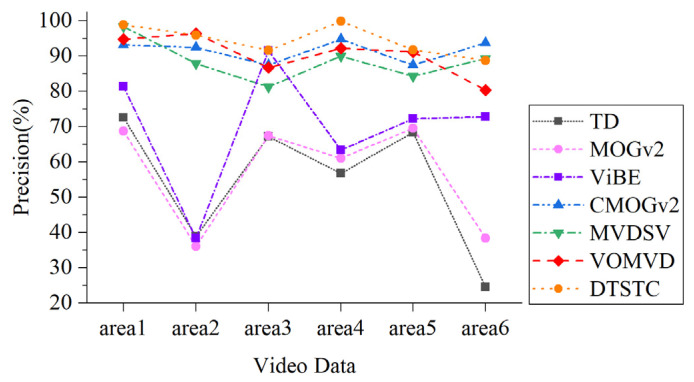
Precision comparison chart.

**Figure 11 sensors-23-05771-f011:**
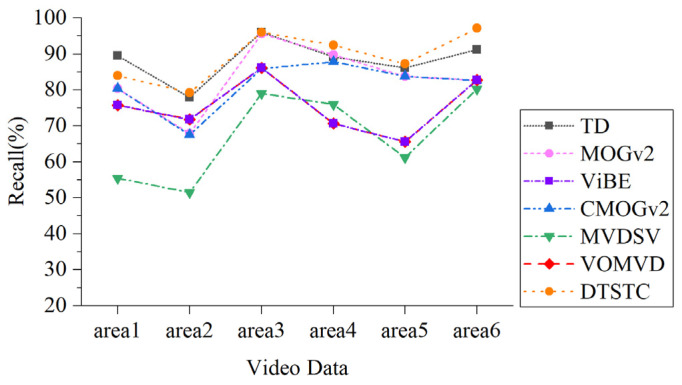
Recall rates comparison chart.

**Figure 12 sensors-23-05771-f012:**
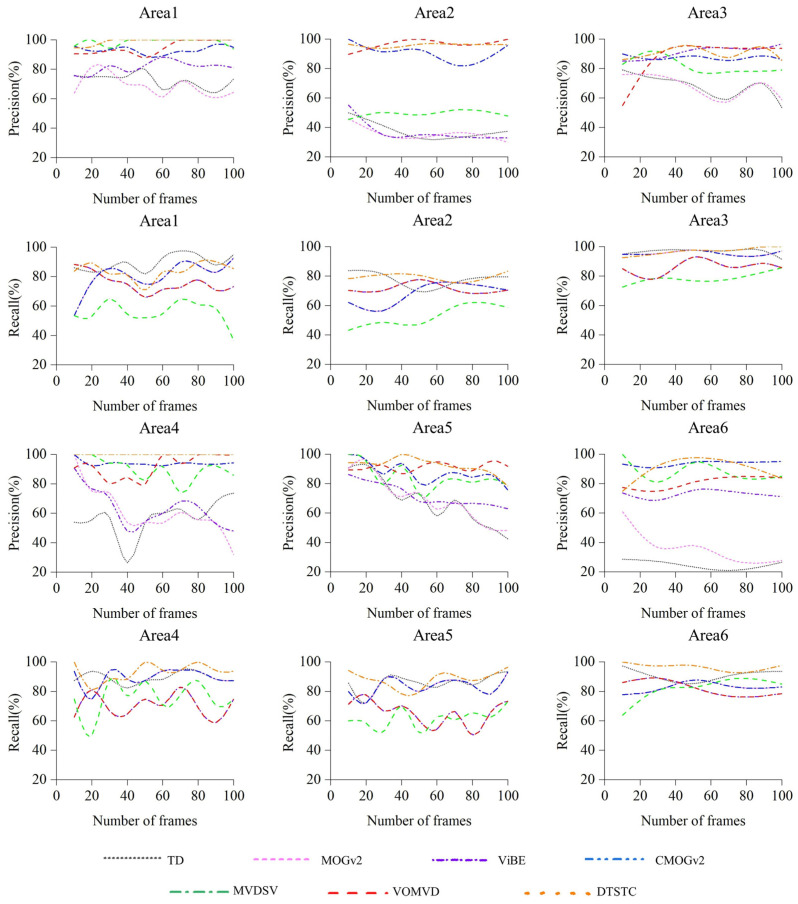
Precision and recall stability assessment results.

**Figure 13 sensors-23-05771-f013:**
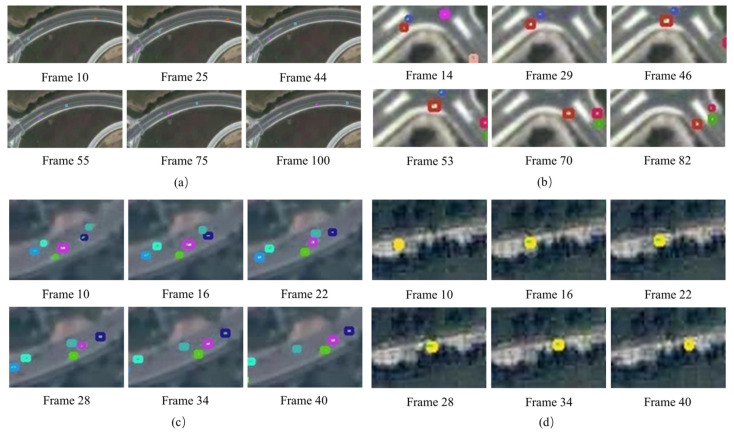
Tracking results in complex situations. The figure (**a**–**d**) depict the tracking results for each of the four different scenarios. The different color boxes in the figure represent different tracking targets.

**Table 1 sensors-23-05771-t001:** Results of different algorithms for vehicle detection.

Test Data		Area 1	Area 2	Area 3	Area 4	Area 5	Area 6
TD	*P* (%)	72.56	38.81	67.21	56.72	68.35	24.52
	*r* (%)	**89.45**	77.88	96.01	89.21	86.08	91.22
	*F* (%)	79.90	51.65	78.80	68.63	74.73	39.49
MOGv2	*P* (%)	68.70	36.06	67.43	61.13	69.61	39.40
	*r* (%)	80.40	67.81	95.63	89.70	83.70	82.67
	*F* (%)	73.63	46.52	78.85	71.31	74.72	51.24
ViBE	*P* (%)	81.47	38.33	91.59	63.38	72.24	72.85
	*r* (%)	75.78	71.84	86.16	70.66	65.61	82.74
	*F* (%)	78.23	49.42	88.74	66.02	68.52	77.36
CMOGv2	*P* (%)	93.17	92.50	87.51	94.77	87.44	**93.76**
	*r* (%)	80.40	67.48	85.88	87.79	83.70	82.67
	*F* (%)	85.88	77.63	86.79	91.04	85.16	87.85
MVDSV	*P* (%)	98.37	87.94	81.37	90.08	84.41	89.40
	*r* (%)	55.43	51.48	79.03	75.93	61.77	80.28
	*F* (%)	70.59	64.67	80.02	81.57	71.05	83.96
VOMVD	*P* (%)	94.80	**96.45**	86.82	92.21	91.26	80.43
	*r* (%)	75.78	71.83	86.17	70.66	65.61	82.73
	*F* (%)	84.00	82.31	85.82	79.65	75.99	81.35
DTSTC	*P* (%)	**98.86**	96.01	**91.69**	**99.99**	**91.84**	88.75
	*r* (%)	84.00	**79.24**	**96.05**	**92.44**	**87.26**	**97.15**
	*F* (%)	**90.03**	**86.80**	**93.28**	**95.96**	**89.22**	**92.47**

Bold text in the table indicates the best value and underlined text indicates the second best value.

**Table 2 sensors-23-05771-t002:** Tracking performance evaluation results.

Test Data	Method	*IDFN*	*IDFP*	*IDTP*	*IDS*	*MOTA* (%)	*IDF*_1_ (%)
Area 1	CMOGv2_D	100	6	329	4	74.71	86.13
	C_SORT	67	2	366	3	83.44	91.38
	DTSTC	**62**	**0**	**373**	**1**	**85.52**	**92.33**
Area 2	CMOGv2_D	117	**15**	239	9	61.99	78.36
	C_SORT	121	18	232	3	61.72	76.94
	DTSTC	**111**	**15**	**245**	**2**	**65.49**	**79.55**
Area 3	CMOGv2_D	44	0	75	3	60.50	77.32
	C_SORT	**0**	**0**	**119**	**0**	**100.00**	**100.00**
	DTSTC	**0**	**0**	**119**	**0**	**100.00**	**100.00**
Area 4	CMOGv2_D	109	33	241	12	59.79	77.24
	C_SORT	**36**	22	325	2	84.33	91.80
	DTSTC	**36**	**2**	**345**	**2**	**89.56**	**94.78**
Area 5	CMOGv2_D	218	6	166	12	39.49	59.71
	C_SORT	**28**	**12**	**350**	**4**	**88.72**	**95.49**
	DTSTC	**28**	**12**	**350**	**4**	**88.72**	**95.49**
Area 6	CMOGv2_D	77	36	137	8	51.60	70.80
	C_SORT	**29**	34	187	7	72.00	85.58
	DTSTC	**29**	**16**	**205**	**5**	**80.00**	**90.11**

Bold in the table indicates the optimal value.

## Data Availability

The data presented in this study are available on reasonable request from the corresponding author.

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
