# Peer review of "Satellite Video Moving Vehicle Detection and Tracking Based on Spatiotemporal Characteristics"

_sensors, 2023, doi:10.3390/s23125771_

Round 1

Reviewer 1 Report

Review report:
•    A brief summary (one short paragraph) outlining the aim of the paper, its main contributions and strengths

Paper presents a method for detecting moving vehicles from satellite videos based on the constraints from spatiotemporal characteristics, fusing road masks from the spatial domain with motion heat maps from the temporal domain. In tests proposed method outperformed the traditional method in constructing constraints, correct detection rate, false detection rate, and missed detection rate.

•    General concept comments
Main observations are:
- bibliography contains 32 references, 5 older than 10 years, 10 references from last 10 years and 17 from last 5 years.
- English language should be improved

•    Specific comments referring to line numbers, tables or figures that point out inaccuracies within the text or sentences that are unclear.
- in abstract, line 11 "However, existing methods for constructing road constraints suffer, from poor stability, high arithmetic performance, leakage, and error detection." High aritmetic performance is an advantage, maybe you maent "low arithmetic performance"
- in 3.3. Inter-frame vehicle association, line 277, equation 9 is not referenced in text
- in 3.3. Inter-frame vehicle association, line 294, equation 10 is not referenced in text
- in 4. Results, 4.1 Test Data, - a description of the six datasets selected for testing should be added
- a name should be given to the proposed method and used instead of "ours"
- in 4.2 Evaluation Indicators, line 322 "F-values (F)" should be F1 (or F1 score)
- in 4.5 Moving vehicle tracking test, results from table 2 should bedescribed as percents - increased values from the other 2 methods CMOGv2+Ours and SORT

•    Is the manuscript clear, relevant for the field and presented in a well-structured manner?
Manuscript is relevant for the field and well presented.

•    Are the cited references mostly recent publications (within the last 5 years) and relevant? Does it include an excessive number of self-citations?
Cited references are new, out of 32 references, 5 are older than 10 years, 10 references are from last 10 years and 17 are from last 5 years.

•    Is the manuscript scientifically sound and is the experimental design appropriate to test the hypothesis?
The experiment design is appropriate to test the hypothesis.

•    Are the manuscript’s results reproducible based on the details given in the methods section?
More information should be added in order for the results to be reproducible.

English language could be improved.

Author Response

Dear Reviewers:

Thank you very much for your suggested changes to manuscript: sensors-2410492. Your suggestions are pertinent and correct and are exactly what we lack in our trials and writing.

We have revised the whole manuscript against your suggestions line by line. The specific changes have been marked in the original manuscript and are explained in detail in the annex. Please take note of them.

Thank you again for your contribution to the manuscript.

Reviewer 2 Report

This paper presents Satellite Video Moving Vehicle Detection and Tracking methods by combining both time-domain and spatial-domain information. They used data association method and employed some machine learning and deep learning algorithms including KNN and ResNet34 respectively.

Overall, the idea and topic of this manuscript are interesting and have some potential significance. However, the overall quality of this manuscript is limited.

The originality of this manuscript is very low, just to propose of using heatmap instead of road map for low memory and adopted algorithms are used to extract features. Results are compared with three algorithms including TD, MOGv2 nad ViBE, which are very old conventional methods were published in 1999, 2002 and 2009 respectively.

Authors must compare their results with contemporary methods.

 AS mentioned, authors considered heat map because of low memory footprint, so need to conduct experiments to show how much memory space are saved. 

List of references are also very old.

Author Response

(The authors gave the same response as above.)

Reviewer 3 Report

Great paper, very good experiments.

Charts can be improved, most of them are pixelated.

The process of creating the street mask can be improved.

Give a name to your proposed algorithm! This way you do not need to write “our algorithm” over and over again.

Repeating the tests with a dataset different from the one used to tune the algorithm, provides more credible results. But overall a very sound testing process otherwise.

Check time forms in your work.

A few corrections needed

Author Response

(The authors gave the same response as above.)

Round 2

Reviewer 1 Report

Thank you very much for your effort, all observations were addressed and all recommendations have been implemented.

English language was improved in the new version.

Reviewer 2 Report

In the revised manuscript, the authors have addressed all the comments and I think, the manuscript is ready for publication.